# More is not enough: High quantity and high quality antenatal care are both needed to prevent low birthweight in South Asia

**Sumanta Neupane**[1], **Samuel Scott**[1]*, **Ellen Piwoz**[2], **Sunny S. Kim**[3], **Purnima Menon**[1], **Phuong Hong Nguyen**[3]

**1** International Food Policy Research Institute, New Delhi, India, **2** Independent Researcher, Annapolis, Maryland, United States of America, **3** International Food Policy Research Institute, Washington DC, United States of America

* samuel.scott@cgiar.org

**Data Availability Statement:** All data underlying the study findings have been uploaded as supporting information.

## Abstract

Antenatal care (ANC) is an opportunity to receive interventions that can prevent low birth weight (LBW). We sought to 1) estimate LBW prevalence and burden in South Asia, 2) describe the number of ANC visits (quantity) and interventions received (quality), and 3) explore associations between ANC quantity, quality and LBW. We used Demographic and Health Survey (DHS) data from Afghanistan (2015), Bangladesh (2018), India (2016), Nepal (2016), Pakistan (2018) and Sri Lanka (2016) (n = 146,284 children <5y). Women were categorized as follows: 1) low quantity (<4 ANC visits) and low quality (<5 of 10 interventions received during ANC), 2) low quantity and high quality ($\geq$5 of 10 interventions), 3) high quantity ($\geq$4 visits) and low quality, 4) high quantity and high quality. We used fixed effect logistic regressions to examine associations between ANC quality/quantity and LBW (<2500 grams). LBW prevalence was highest in Pakistan (23%) and India (18%), with India accounting for two-thirds of the regional burden. Only 8% of women in Afghanistan received high quantity and high quality ANC, compared to 42–46% in Bangladesh, India, and Pakistan, 65% in Nepal and 92% in Sri Lanka. Compared to the low quantity/quality reference group, children of women with high quantity/quality ANC had lower odds of LBW in India (Adjusted Odds Ratio 0.84, 95% CI 0.78–0.89), Nepal (0.57, 0.35–0.94), Pakistan (0.45, 0.23–0.86), and Sri Lanka (0.73, 0.57–0.92). Low quantity but high quality ANC was protective in India (0.90, 0.84–0.96), Afghanistan (0.53, 0.27–1.05) and Pakistan (0.49, 0.23–1.05). High quantity but low quality ANC was protective in Sri Lanka (0.76, 0.61–0.93). Neither frequent ANC without appropriate interventions nor infrequent ANC with appropriate interventions are sufficient to prevent LBW in most South Asian countries, though quality may be more important than quantity. Consistent measurement of interventions during ANC is needed.

**Funding:** This work was supported by the Bill and Melinda Gates Foundation through the DataDENT initiative (grant number OPP1174256). The funders had no role in study design, data collection and analysis, decision to publish, or preparation of the manuscript.

**Competing interests:** The authors have declared that no competing interests exist.

**Abbreviations:** ANC, Antenatal Care; CI, Confidence Interval; DHS, Demographic and Health Survey; IFA, Iron and Folic Acid; LBW, Low Birth Weight; MICE, Multivariate Imputation by Chained Equations; OR, Odds Ratio.

## Introduction

The World Health Organization (WHO) recommends a minimum four antenatal care (ANC) contacts during pregnancy–and more recently, in 2016, at least eight ANC contacts [1]–so that women can receive all recommended interventions to achieve optimal pregnancy and birth outcomes. The overarching goal of ANC should be to ensure that women have a positive pregnancy experience. While having the recommended number of ANC contacts is important, a woman who attends all recommended ANC visits may not receive optimal quality of care. In an analysis of 10 countries across Latin America, Africa and Asia (Nepal), among women who started ANC early in their pregnancy and completed at least four visits, the percentage of women who received all six common interventions (blood pressure measured, urine sample taken, blood sample taken, tetanus protection, iron supplementation, and receipt of information on potential complications) was less than 50% in half of the countries [2]. Little is known about how ANC quantity and quality together relate to birth outcomes that contribute to neonatal mortality such as low birthweight (LBW).

LBW (defined as birthweight< 2500g), a direct result of preterm birth (gestational age <37 completed weeks) and/or intrauterine growth restriction, accounts for approximately 80% of neonatal deaths globally [3]. LBW is not only a major predictor of neonatal mortality and early childhood morbidity, but also of noncommunicable diseases later in life [4]. Nearly 15% of children globally are born with LBW [5]. In South Asia, 26% of children are born with LBW [5], the highest of any region. Between 2000 and 2015, the prevalence of LBW in South Asia declined by 5.9 percentage points (pp) with an average annual rate of reduction (AARR) of 1.4 pp, slighter faster than the global AARR of 1.2 pp/year [5]. However, the pace of reduction is not sufficient to achieve the World Health Assembly target of a 30% reduction in LBW between 2012 and 2025 [5, 6], and accelerating the reduction in South Asia is critical given the large LBW burden in the region.

Interventions delivered during ANC can prevent and address risk factors for LBW such as maternal hypertension, underweight, anemia, and poor diet quality [4, 7–11]. Previous studies have explored the independent contributions of either the number of ANC visits or of interventions received during ANC to the risk of LBW. Number of ANC visits is positively associated with LBW [12–14] as are interventions during ANC such as iron folic acid supplementation [15] and nutrition education [16]. Only one study to our knowledge, an analysis linking household and health facility data from Malawi, has considered quality adjusted ANC visits in relation to LBW; the authors concluded that delivering nutrition interventions within the existing level of coverage would decrease population prevalence of LBW from 13.7% to 10.8% [17]. We are not aware of any studies that have examined the relative contributions of ANC quantity and ANC quality to the risk of LBW using household survey data only, across multiple countries. In this paper, we examine the independent and combined contributions of ANC quantity and quality on child birthweight in six South Asian countries. Specifically, we aim to answer three research questions: 1) What is the national and subnational prevalence of LBW in South Asian countries? 2) How do women's ANC experiences (quantity and quality) compare across South Asian countries? and 3) What is the association between ANC quantity/quality and LBW?

## Methods

### Data sources

We used household data from the most recent round of Demographic and Health Survey (DHS) conducted in six South Asian countries: Afghanistan (2015) [18], Bangladesh (2018)

[19], India (2016) [20], Nepal (2016) [21], Pakistan (2018) [22], and Sri Lanka (2016) [23]. The National Family Health Survey (NFHS) in India is equivalent to DHS. All DHS collected information on children born in the past five years except in Bangladesh (in the past three years).

## Measures

The primary outcome was LBW which was defined as a birthweight < 2500 grams. Birthweight data were collected either from health record cards or from mothers' recall (both in grams): 47% recorded versus 53% recalled in Afghanistan; 4% vs. 95% in Bangladesh; 55% vs. 45% in India; 20% vs. 80% in Nepal; 13% vs. 87% in Pakistan; 100% recorded in Sri Lanka. If recorded birthweight was not available, we used recalled birthweight. Data on birthweight, from either record or recall, was missing for a subset of children in all countries, with the highest percentage of children with missing birthweight in Afghanistan (86%), followed by Pakistan (80%), Bangladesh (54%), Nepal (35%), India (23%) and Sri Lanka (2%). The LBW burden was the product of LBW prevalence, and the number of children born during the survey year, using United Nations Population Division numbers [24].

The key independent variables were ANC quantity and quality. ANC quantity was assessed based on the number of ANC visits completed by a woman during her pregnancy with her youngest child. We defined at least four ANC visits as "high quantity" and less than four ANC visits as "low quantity". We chose the cutoff of four visits even though the recent WHO recommendation [1] is at least eight visits because the proportion of women who reported having eight or more ANC visits was too low to be able to conduct the regression analysis and also because the surveys were too early for countries to implement the WHO's 2016 recommendation.

To define ANC quality, we considered the number of interventions received during any ANC visit during the woman's pregnancy with her youngest child and included 12 interventions with available data: 1) weight assessment, 2) blood pressure measurement, 3) blood sample collected, 4) urine sample collected, 5) consumed 100+ iron folic acid (IFA) supplements, 6) consumed calcium supplements, 7) ultrasound conducted, 8) received counseling on danger signs, 9) received two tetanus shots, 10) received preventive deworming, 11) received food supplements, and 12) received health and nutrition education.

Of these 12 ANC interventions, data were available on seven interventions in Afghanistan, Bangladesh, Nepal, and Pakistan, on ten in Sri Lanka and on 11 in India. The total number of ANC interventions varied because of country-specific ANC guidelines and survey question differences. To enable cross-country comparison, we standardized the number of interventions received on a scale of ten (we divided the proportion of interventions received by the number of interventions received and multiplied by ten). Women with a standardized score of five or more out of ten were categorized as having "high quality" ANC and women with a standardized score below five had "low quality" ANC. Finally, we grouped women into four quantity/quality categories based on their ANC experience: 1) low quantity and low quality, 2) low quantity and high quality, 3) high quantity and low quality, and 4) high quantity and high quality.

Confounding variables were considered at the maternal level (women's age, education, body mass index [BMI] calculated by dividing weight in kilograms by squared height in meters, and underweight or BMI <18.5 kg/m$^2$), child level (child's birth order and sex), and household level (household's wealth index [derived using principal component analysis on household durable assets, building material of dwelling and sanitation facilities] and place of residence [urban or rural/estate]).

### Addressing missing birthweight data

Given the high percentage of missing birthweight data, we also conducted analysis using a subjective report of the child's birth size as perceived by the mother (very small, small, normal, large, and very large). A multi-country analysis of DHS surveys observed that perceived birth size is a good proxy for birth weight [25]. We categorized very small and small as low birth size as others have previously done [8]. Perceived birth size data was available for most children. The total number of youngest children born in the past five years was 234,931; of those, birthweight was available for 163,973 children and perceived birth size was available for 213,684 children (**S1 Table**). Perceived birth size was not asked in Bangladesh or Sri Lanka.

Samples with and without data may systematically vary from each other leading to biased estimates. For each survey, we compared the selected characteristics among samples with and without birthweight data using a Student t-test for continuous variables or Chi-square test for categorical variables and included variables that were statistically significant ($p < 0.05$) between two groups as covariates in the regression models.

### Statistical analysis

Descriptive analysis was used for sample characteristics of the six country datasets. We used bar charts to show the prevalence and burden of LBW and used maps to illustrate the subnational variation in its prevalence. To explore the association between LBW and the combined ANC quantity/quality indicator, we ran logistic regression models separately for each country. The LBW outcome was defined as recorded birthweight < 2500 grams or, if recorded birthweight was not available, recalled birthweight <2500 grams. We implemented two additional variations of the models as robustness checks: 1) LBW defined as low/very low perceived birth size, 2) LBW defined as recorded/recalled birthweight <2500 grams or low/very low perceived birth size (used if birthweight data were missing). We included state or division (depending on the country) as a fixed effect in our models to control for variation in unmeasured factors at the state level that may influence LBW. To account for the multistage cluster sampling design and varying probability of being selected as a sample, we applied survey data analysis procedures in Stata 15.0. We report the adjusted odds ratio (AOR) and 95% confidence intervals (CI) and used significance levels at $p < 0.05$, $p < 0.01$ and $p < 0.001$.

In Nepal and Pakistan, weight and height measurements were missing for more than 50% of the women, thus we imputed BMI using multivariate imputation by chained equations (MICE). This generated multiple imputed datasets for the BMI variable, using predictive mean matching algorithms. We included all predictors and the outcome in the imputed model as recommended [26]. In total, ten imputed datasets were created. The modeled coefficients were then pooled across the imputed sets. Missing BMI values were replaced with BMI predicted from MICE prior to running regression models.

## Results

### Sample characteristics

Samples of women from different countries varied in terms of mean age (25 years in Bangladesh to 31 years in Sri Lanka), having no education (<1% in Sri Lanka to 83% in Afghanistan), underweight (11% in Pakistan and Sri Lanka to 25% in India), and having at least three births (25% in Sri Lanka to 68% in Afghanistan) (**Table 1**). Samples also varied in child and household characteristics, with a range of mean child age (17 months in Bangladesh to 29 months in Sri Lanka), and percentage of households from rural areas (44% in Nepal to 80% in Sri Lanka).

**Table 1. Descriptive characteristics of women (15–49 years), youngest children (0–59 months) and households, by country in South Asia.**

|  | Afghanistan 2015 | Bangladesh 2018 | India 2016 | Nepal 2016 | Pakistan 2018 | Sri Lanka 2016 |
|---|---|---|---|---|---|---|
| N | 19,689 | 5,012 | 190,709 | 4,006 | 8,286 | 7,040 |
| Woman's age at survey, years | 28.9 | 24.9 | 26.9 | 26.4 | 29.4 | 31.1 |
| Woman's education |  |  |  |  |  |  |
| No education, % | 82.9 | 6.3 | 27.6 | 31.4 | 47.9 | 0.7 |
| Primary, % | 8.1 | 27.6 | 13.4 | 19.4 | 16.3 | 3.5 |
| Secondary, % | 7.3 | 49.0 | 46.9 | 33.6 | 22.2 | 66.1 |
| Higher, % | 1.7 | 17.1 | 12.0 | 15.5 | 13.6 | 29.7 |
| Woman's BMI<18.5 kg/m², % | - | 15.7 | 24.8 | 18.3 | 11.2 | 11.3 |
| Woman's number of births |  |  |  |  |  |  |
| 1, % | 15.2 | 38.2 | 33.6 | 37.5 | 19.9 | 36.2 |
| 2, % | 16.8 | 32.8 | 34.5 | 30.2 | 21.0 | 38.5 |
| ≥3, % | 68.0 | 29.0 | 31.9 | 32.3 | 59.0 | 25.3 |
| Child sex is female, % | 47.8 | 47.8 | 45.6 | 45.0 | 49.3 | 48.2 |
| Child age, months | 21.1 | 16.6 | 25.5 | 26.1 | 21.9 | 28.7 |
| Household residence is rural, % | 76.7 | 73.2 | 70.3 | 44.4 | 66.5 | 80.3[1] |
| Household wealth quintile[2] |  |  |  |  |  |  |
| Poorest, % | 20.5 | 18.8 | 18.9 | 17.0 | 22.1 | 16.7 |
| Second, % | 18.6 | 20.8 | 18.5 | 18.8 | 16.0 | 19.6 |
| Third, % | 18.1 | 19.8 | 19.3 | 20.7 | 18.7 | 20.5 |
| Fourth, % | 19.6 | 21.9 | 21.1 | 21.0 | 21.6 | 21.5 |
| Richest, % | 23.2 | 18.8 | 22.2 | 22.5 | 21.6 | 21.6 |

[1]Also includes estate (Urban = 15.5%, Rural = 80.4%, Estate = 4.1%)

[2]Wealth index, derived separately for countries by running a principal component analysis on housing characteristics and household's possession of durable assets, was divided into wealth quintiles.

## Prevalence and burden of low birthweight

LBW prevalence was highest in Pakistan (23%), followed by India (18%), Afghanistan (15%), Bangladesh (15%), Sri Lanka (15%), and Nepal (11%) (**Fig 1A**). The burden of LBW in terms of absolute numbers was highest in India (4.27 million or 67% of the total regional burden), followed by Pakistan (1.36 million or 21%) and Bangladesh (450,000 or 7%) (**Fig 1B and 1C**). Afghanistan, Nepal, and Sri Lanka combined accounted for only 5% of the total regional burden. Within countries, we observed subnational variation in LBW prevalence (**S1 Fig and S2 Table**). The highest subnational variation was in Afghanistan (0% to 62% within districts), followed by Pakistan (6% to 25%), India (8% to 23%), Sri Lanka (9% to 21%), Bangladesh (9% to 20%) and Nepal (8% to 13%).

## Women's ANC experience

The percentage of women who had at least four ANC visits during pregnancy with their youngest child ranged from 12% in Afghanistan to 88% in Sri Lanka (**Table 2**). The percentage of women with at least 8 ANC visits ranged from 2% in Afghanistan to 58% in Sri Lanka (**Table 2**). More than one-third (38%) of women did not have any ANC visits in Afghanistan.

Out of 12 ANC quality indicators, only four (blood pressure measurement, blood sample collected, urine sample collected and taken 100+ IFA supplements) were available in all countries (**Table 2**). Data on calcium supplementation was available only in Sri Lanka; food supplementation and health/nutrition education in two countries; counselling on danger signs in

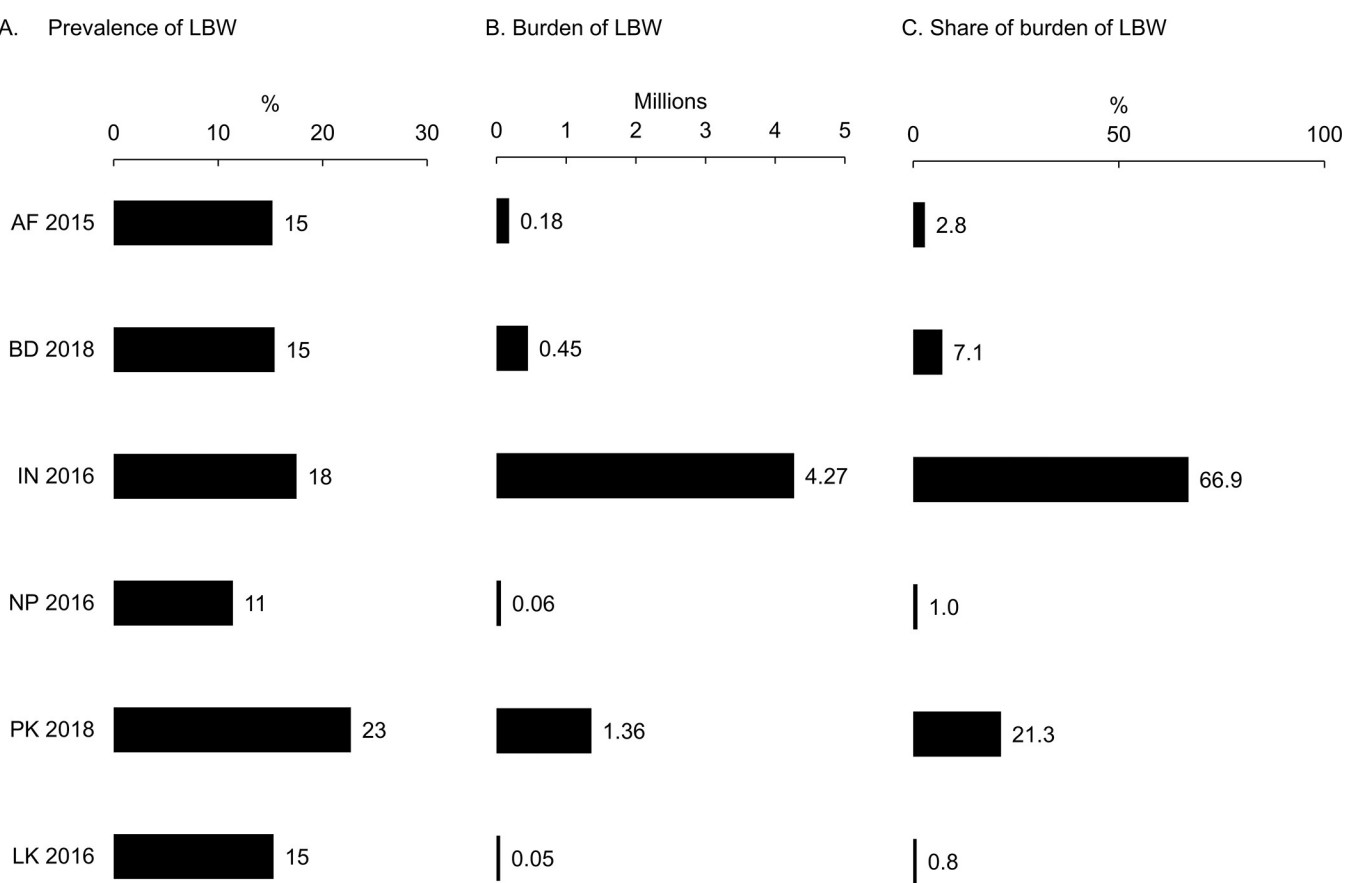

**Fig 1. Prevalence and burden of low birthweight in South Asia.** Data are for the youngest child born in the past five years. The number of children born with LBW is the product of the prevalence of LBW (black bars and number of births in the same year using United Nations Population Division estimates (Our World in Data website. Accessed March 25, 2021. https://ourworldindata.org/grapher/births-and-deaths-projected-to-2100). The share of the burden is the burden for each country divided by the total regional burden (e.g., for India, (4.27/6.37)*100 = 66.9%). AF = Afghanistan; BD = Bangladesh; IN = India; NP = Nepal; PK = Pakistan; LK = Sri Lanka.

four countries; women's weight assessment and ultrasound conducted in three countries; two tetanus shots and preventive deworming in five countries. More than half of women in all countries had their blood pressure measured, blood sample collected, and urine sample collected except in Afghanistan (49%, 19%, and 25% respectively) (**Table 2**). More than half of women had their ultrasound conducted in Bangladesh (74%), India (68%) and Sri Lanka (89%). In all countries, the percentage of women who took 100+ IFA supplements during pregnancy was below 50% except in Nepal (65%) and Sri Lanka (96%). The percentage of women who reported receiving preventive deworming was low in Afghanistan (3%), India (18%), Pakistan (2%). The percentage of women who reported getting both tetanus shots ranged between 34% in Afghanistan and 83% in India. Receipt of food supplements during pregnancy was nearly twice as common in Sri Lanka (94%) as India (53%). Health/nutrition education was 39% in India and 61% in Pakistan. Based on the standardized score cut-off of five, the ANC quality score was highest in Sri Lanka (9.0 out of 10) and lowest in Afghanistan (2.3 out of 10).

When we grouped women into four combined quantity/quality categories, only 8% of women in Afghanistan received both high quantity and high quality ANC, compared to 42–46% in Bangladesh, India and Pakistan, 65% in Nepal, and 92% in Sri Lanka (**Fig 2**).

**Table 2. Quantity and quality of antenatal care during women's pregnancy with their youngest child under five years old in South Asian countries.**

| | Afghanistan 2015 | Bangladesh 2018 | India 2016 | Nepal 2016 | Pakistan 2018 | Sri Lanka 2016 |
|---|---|---|---|---|---|---|
| ANC quantity (No. ANC visits) | | | | | | |
| 0, % | 38.1 | 8.0 | 16.4 | 5.9 | 12.2 | 3.5 |
| 1, % | 19.4 | 25.1 | 16.0 | 33.2 | 22.2 | 4.5 |
| 2, % | 17.7 | 16.4 | 12.4 | 8.0 | 14.3 | 1.6 |
| 3, % | 12.5 | 15.5 | 13.4 | 13.2 | 13.4 | 2.3 |
| $\geq$ 4, % | 12.3 | 35.0 | 41.8 | 39.7 | 38.0 | 88.2 |
| $\geq$ 8, % | 2.3 | 11.1 | 18.2 | 8.6 | 14.0 | 57.8 |
| ANC quality items | | | | | | |
| Weight assessed, % | - | 81.1 | 75.6 | - | - | 96.4 |
| Blood pressure measured, % | 48.8 | 85.8 | 74.7 | 85.9 | 78.5 | 97.8 |
| Blood sample collected, % | 18.5 | 60.4 | 73.0 | 62.4 | 61.8 | 91.3 |
| Urine sample collected, % | 24.5 | 66.4 | 73.5 | 71.6 | 62.2 | 94.3 |
| Taken 100+ IFA suppl[1]., % | 3.5 | 33.5 | 22.1 | 65.0 | 19.7 | 95.6 |
| Taken calcium suppl[2]., % | - | - | - | - | - | 94.6 |
| Ultrasound conducted, % | - | 73.8 | 68.4 | - | - | 89.0 |
| Counseling on danger signs, % | 36.5 | 36.5 | 53.4 | 74.0 | - | - |
| Received 2 tetanus shots, % | 34.2 | - | 83.5 | 66.1 | 63.4 | 45.5 |
| Received preventive deworming, % | 3.2 | - | 18.2 | 70.2 | 1.8 | 94.9 |
| Received food suppl., % | - | - | 52.5 | - | - | 93.8 |
| Health/nutrition education, % | - | - | 39.2 | - | 61.2 | - |
| ANC quality score[3] (out of 10) | 2.3 | 6.2 | 5.8 | 7.1 | 5.0 | 9.0 |

[1] In Sri Lanka DHS did not ask the number of days IFA received, so this intervention in Sri Lanka was whether received IFA

[2] Calcium supplementation is not part of recommended nutrition interventions for pregnant women in Nepal

[3] ANC quality score was calculated by adding the number of ANC interventions received by women in each country and later standardized to a scale of 10.

## Association between ANC quantity/quality categories and low birthweight

In the models with LBW defined according to birthweight data, compared to women with low ANC quantity and quality during pregnancy, women who had received both high quantity and quality ANC had lower odds of giving birth to a LBW child in India (AOR 0.84, 95% CI 0.78–0.89), Nepal (0.57, 0.35–0.94), Pakistan (0.45, 0.23–0.86), and Sri Lanka (0.73, 0.57–0.92) (**Fig 3** and **S3 Table**). Low quantity but high quality ANC was significantly associated with reduced odds of LBW in India (0.90, 0.84–0.96), and non-significantly in Afghanistan (0.53, 0.27–1.05) and Pakistan (0.49, 0.23–1.05). High quantity but low quality ANC was associated with reduced odds of LBW in Sri Lanka (0.76, 0.61–0.93). Maternal higher education (in 4 of 6 countries), maternal not being underweight (in 2 of 5 countries with data) and coming from relatively wealthy households (in 2 of 6 countries) were also protective against LBW (**S3 Table**).

## Robustness checks

Characteristics of women, children, and households varied significantly among samples with and without birthweight (**S4 Table**). Women with higher education, gave birth to a first child, living in urban area and from wealthier households were more likely to have birthweight data. When we ran our regression models using LBW defined as very low or low perceived birth size, results were similar to results from the primary analysis using birthweight data in 3 of 4 countries with available data: Afghanistan, India and Nepal (**S5 Table**). When we defined

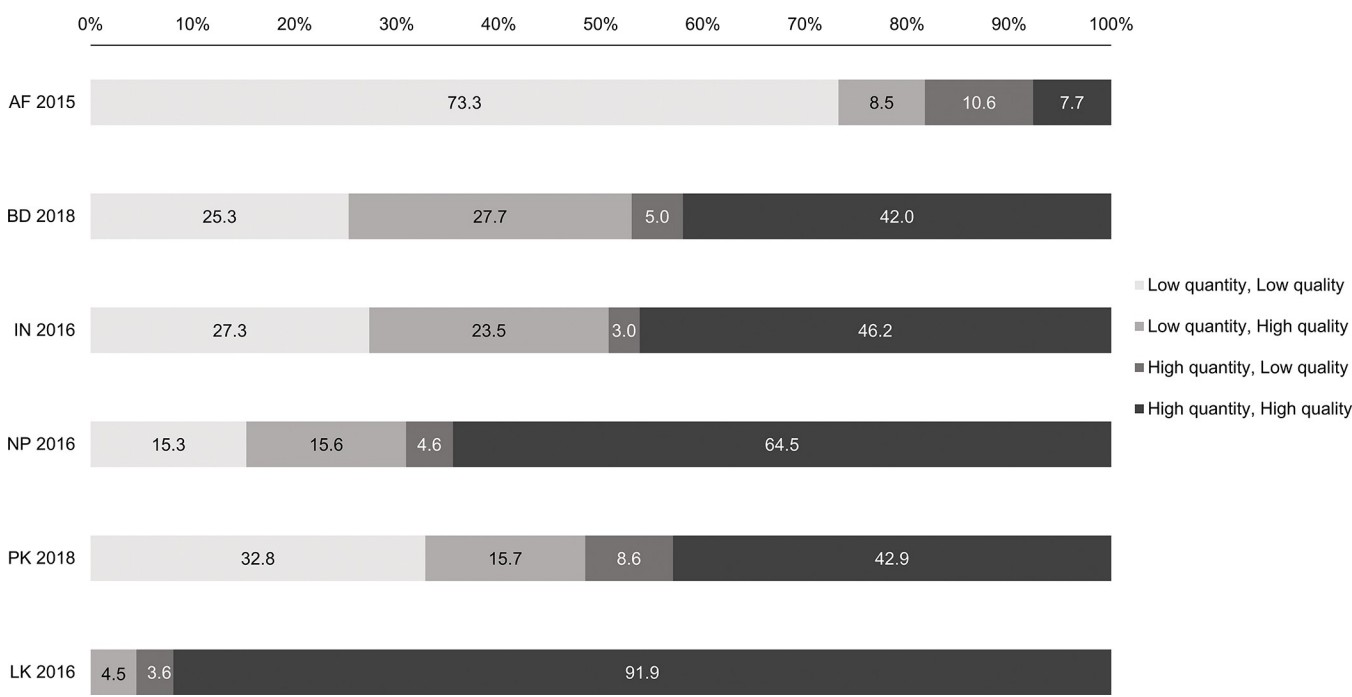

**Fig 2. Percentage of women in each ANC quantity and quality group by country in South Asia.** High ANC quality was defined as a woman with a standardized ANC score more than 5. High ANC quantity was defined as a woman having at least 4 ANC visits. AF = Afghanistan; BG = Bangladesh; IN = India; NP = Nepal; PK = Pakistan; LK = Sri Lanka.

LBW using either birthweight or perceived birth size data, results were similar to those found in the main analysis that used only birthweight data (**S6 Table**). Additionally, we compared birthweight models using recorded and recalled birthweight as separate outcomes, and again these yielded findings similar to the primary analysis that used recorded birthweight if available and recalled birthweight if recorded birthweight was not available (**S7 Table**). Finally, given the large sample size in India, we examined how the effect size changes if we changed the outcome to very/extremely LBW (<2000 grams) instead of LBW (<2500 grams). Approximately 4% of the sample (5972 individuals) were very/extremely LBW (**S8 Table**). The protective effect of high quantity and high quality ANC was slightly stronger for very/extremely LBW (0.77, 0.67–0.89) (**S9 Table**) compared to LBW (0.84, 0.78–0.89) (**Fig 3**).

## Discussion

### Summary of findings

Using nationally representative household samples from six South Asian countries, we examined the relationship between women's ANC experience and child LBW. We observed that LBW ranged from 11% (Nepal) to 23% (Pakistan) in the region, affecting approximately 6.37 million newborns annually. India accounts for two-thirds of the regional burden. Women did not attend ANC frequently enough, with more than half of women in five out of six countries attending fewer than four ANC visits during their most recent pregnancy. Aside from Sri Lanka, ANC quality was also generally low, with few women receiving all recommended interventions to promote optimal pregnancy and birth outcomes. In four of six countries, compared to women with low quantity and quality ANC, those with high quantity and quality ANC had significantly lower odds of giving birth to a LBW child, after adjusting for a range of

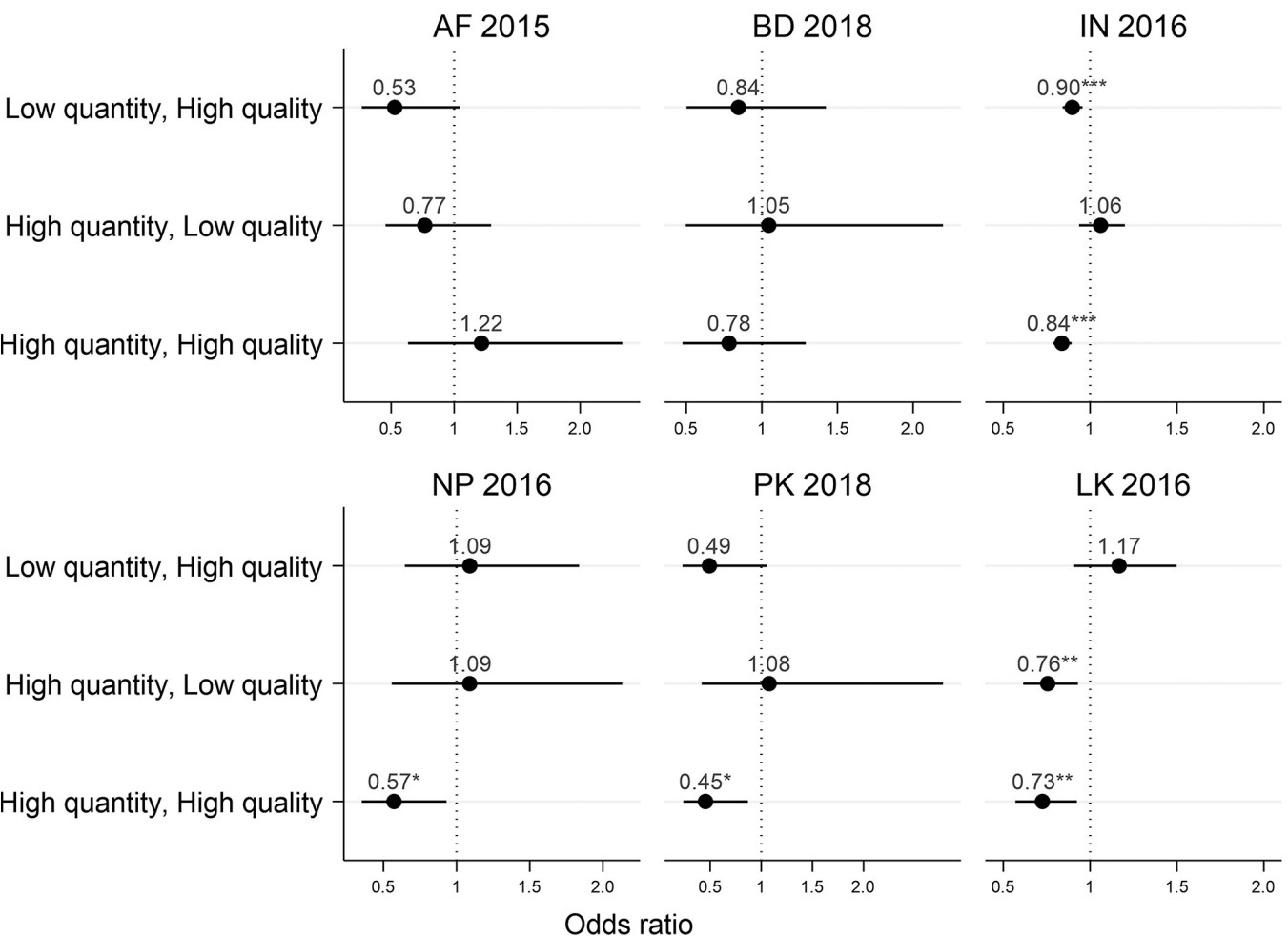

**Fig 3. Adjusted odds of low birthweight by ANC quantity and quality group, in South Asian countries.** The reference group for all models is low quantity, low quality. Models were adjusted for women's age, education and BMI, child's sex and birth order and place of residence and wealth quintile. Women's height and weight was not measured in Afghanistan DHS, as a result we were not able to calculate the BMI. Models also included state or division fixed effects. High ANC quality was defined as a woman with a standardized ANC score more than 5. High ANC quantity was defined as a woman having at least 4 ANC visits. In Sri Lanka, high ANC quality was defined as receiving 10 out of 10 interventions and high ANC quantity was defined as at least 8 ANC visits due to limited variation in ANC quantity and quality. AF = Afghanistan; BD = Bangladesh; IN = India; NP = Nepal; PK = Pakistan; LK = Sri Lanka. ***p<0.001, **p<0.01 *p<0.05.

potential confounding individual, household, and state-level variables. These findings imply that children are most protected against LBW when their mothers attend enough ANC visits and receive most of the recommended interventions during pregnancy. Finally, our results in four of six countries are suggestive of the idea that few high quality ANC visits may be more protective against LBW than many low quality ANC visits, implying that ANC guidelines should emphasize quality–receiving appropriate interventions at the appropriate time during pregnancy–over recommending more ANC visits without attention to delivery of effective interventions.

## Comparison with previous literature

Most studies consider ANC quantity and quality separately in their risk factor analysis of LBW. In terms of quantity, better birth outcomes are associated with more ANC visits. A hospital-based matched case control study from rural Pakistan reported that the risk of LBW

doubled when women had fewer than two ANC visits compared to at least four ANC visits [27]. Similarly, an analysis of DHS data from 18 countries observed that, relative to women with at least four ANC visits, those with fewer than four visits were more likely to have LBW children (OR = 1.5) [12]. A risk factor analysis of LBW in 6 countries including India and Pakistan observed higher odds of child LBW for women reporting 1–3 ANC visits (OR = 1.68) compared to women who reported at least four ANC visits [13].

Studies that focus on ANC quality also report a protective effect of receiving recommended interventions during pregnancy on child LBW. A meta-analysis showed that routine daily IFA consumption during pregnancy reduced the prevalence of LBW by 20% [15]. A 3-month randomized controlled trial among 300 pregnant women in urban Bangladesh tested a nutrition education intervention during ANC focused on promoting consumption of supplemental food; the researchers observed that LBW was 94% lower in the intervention compared to the control group who only received routine health services [16].

We are aware of one study, an analysis geographically linking household survey and health facility data in Malawi, that considered both quantity and quality of ANC together in relation to child LBW [17]. The authors report that one additional quality-adjusted ANC visit (a combined score of number of the number of ANC visits, IFA supplementation, nutrition counseling and breastfeeding counseling) reduced the risk of LBW by 13%.

## Discussion on current findings

Our results indicate that coverage of clinical screenings (assessment of weight, blood pressure, urine, ultrasound, etc.) was higher than coverage of nutrition-related interventions involving supplements or counseling. Assuming the availability of equipment, clinical screenings are relatively easy to administer. Assuring consumption of micronutrient supplements require an adequate supply, instructions around use and benefits, family support [28], and monitoring to ensure compliance. Counseling requires a trained health care provider with the time and capacity to effectively convey relevant messages and relies on the mother to listen to and retain the messages. These interventions need to be prioritized within ANC. Building the capacity of health care providers is likely to lead to better quality care which will, in turn, generate demand and increase the frequency of contacts.

Reduced LBW was significantly associated with high ANC quantity alone in Sri Lanka, where we defined quality differently from other countries and high ANC quality alone in India. Afghanistan and Pakistan showed statistically non-significant but large point estimates for high quality alone having a protective effect. In Sri Lanka, given the overall high quality and quantity in the country, creating contrasting groups to compare in our regression model was a challenge. Women in the "low quality" group still received 8 interventions on an average, and this lack of contrast between groups may partially explain similar odds of LBW between groups. Compared to Sri Lanka, women in India, Afghanistan, and Pakistan experience poorer ANC quality in terms of receiving fewer interventions during their pregnancy. In contexts where ANC quality is relatively poor, increasing the number of visits may not be as protective against LBW as having a few visits and delivering appropriate interventions at those visits.

## Policy implications

Every child should be weighed at birth and that data recorded both for families and in public data systems. Birthweight data was missing for a large proportion of women in all countries, especially Afghanistan (86%) and Pakistan (80%). Measuring and recording weight at birth is a simple action that would help improve programs and outcomes, and the absence of any birthweight data–either recorded or recalled–is striking. Measuring and recording birthweight

would probably also trigger better postnatal follow up for small newborns who are at highest risk.

Comparing ANC quality across countries is complicated by inconsistent measurement. Only four indicators were available in all six countries, and several indicators (weight assessment, calcium supplementation, counseling on danger signs, received food supplements, and health/nutrition education) were available in three or fewer countries. These gaps suggest a lack of agreement on the standard set of ANC quality indicators that should be measured in national health surveys, as others have noted [2], which needs to be rectified.

Adherence to global ANC recommendations in national policies and guidelines varies within the region. Only Afghanistan and Sri Lanka have adjusted their policies to adhere with WHO's recommendation of eight ANC visits [28, 29]. All countries included in our analyses have provisions for all recommended interventions and screening during ANC visits in their policy and program guideline documents aside from calcium supplementation in Nepal [30]. However, our results show that guidelines do not translate into measurement or delivery. Even for the interventions with available data, coverage is low. Given the importance of birthweight for newborn survival and health, greater attention and resourcing is needed to ensure that all women receive recommended interventions at the appropriate time during pregnancy, particularly interventions such as micronutrient supplementation that directly impact birth weight. Evidence demonstrates that improving availability and quality of ANC will improve demand and health service utilization, initiating a positive feedback loop [31].

## Strengths and limitations

Our study has several strengths. First, we used data from nationally representative household surveys in six countries that make up the bulk of the population in South Asia. Second, we explored variations of the LBW outcome, using recorded or recalled birthweight as well as perceived birth size; findings were consistent across these variations. Third, we adjusted for important variables that may confound the association between ANC experience and LBW at individual, household, and state/province levels.

However, our study is not without limitations. First, while we defined ANC quality strictly based on women's report of service receipt to gain insight on their recalled experience during pregnancy at the population level, health facility records of birthweight may be more accurate. While health facility assessments are useful for understanding service availability and readiness (quality of care elements) [32], this data is not available in most countries and it does not provide information on population-based coverage. Second, our high ANC quantity indicator (4 + visits) was based on the previous WHO's recommendation since, when the surveys were conducted, there was not enough time for countries to implement the new recommendations; however, analyses of future survey rounds should also consider the new WHO recommendation (8+ visits). Third, we did not include timing of ANC in our analysis due to sample size constraints (instead of 4 quality/quantity groups, there would be 9 quantity/quantity/timing groups to compare). ANC timing (receiving ANC in the first trimester) is also strongly correlated to ANC quantity, so adding the timing dimension would likely not lead to different inferences.

## Conclusions

More is not enough: both high ANC quantity and quality are needed to protect against LBW. Our results stress the critical importance of simultaneously increasing ANC utilization and improving the quality of ANC services provided to ensure positive birth outcomes. As countries focus on increasing the number of ANC visits to meet updated WHO recommendations,

they need to ensure delivery of appropriate interventions. There are many effective, scalable health and nutrition interventions available to reduce LBW. These interventions are in policy guidance documents but are still not widely delivered. Greater attention should be given to weighing children at birth and systematically measuring ANC service quality as well as quantity in national surveys.

## Supporting information

**S1 Fig. Subnational variation in the prevalence of low birthweight (<2500 grams) in South Asian countries.** Shape file accessible at: https://gadm.org/download_country_v3.html. (DOCX)

**S1 Table. Data availability for birthweight and perceived birth size of children in South Asian countries.**
(DOCX)

**S2 Table. Subnational prevalence of low birthweight, by country in South Asia.**
(DOCX)

**S3 Table. Association between low birthweight and combination of ANC quantity and quality, by country in South Asia.**
(DOCX)

**S4 Table. Comparison of samples with and without birthweight data, by country in South Asia.**
(DOCX)

**S5 Table. Association between perceived birth size of child and combination of ANC quantity and quality, by country in South Asia.**
(DOCX)

**S6 Table. Association between recorded vs.** recalled birthweight and combination of ANC quantity and quality, by country in South Asia.
(DOCX)

**S7 Table. Association between birthweight / birth size and combination of ANC quantity and quality, by country in South Asia.**
(DOCX)

**S8 Table. Prevalence of severity of low birthweight in South Asian countries.**
(DOCX)

**S9 Table. Association between severity of LBW and combination of ANC quantity and quality.**
(DOCX)

**S1 Data. Working dataset for Afghanistan analysis.**
(DTA)

**S2 Data. Working dataset for Bangladesh analysis.**
(DTA)

**S3 Data. Working dataset for India analysis.**
(DTA)

**S4 Data. Working dataset for Nepal analysis.**
(DTA)

**S5 Data. Working dataset for Pakistan analysis.**
(DTA)

**S6 Data. Working dataset for Sri Lanka analysis.**
(DTA)

## Author Contributions

**Conceptualization:** Sumanta Neupane, Samuel Scott, Phuong Hong Nguyen.

**Formal analysis:** Sumanta Neupane, Samuel Scott, Phuong Hong Nguyen.

**Writing – original draft:** Sumanta Neupane, Samuel Scott.

**Writing – review & editing:** Ellen Piwoz, Sunny S. Kim, Purnima Menon, Phuong Hong Nguyen.

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
