## [Decision Letter · Decision Letter 0]

17 Jan 2023

PGPH-D-22-01824

More is not enough: high quantity and high quality antenatal care are both needed to prevent low birthweight in South Asia

Dear Dr. Scott,

Thank you for submitting your manuscript to PLOS Global Public Health. After careful consideration, we feel that it has merit but does not fully meet PLOS Global Public Health’s publication criteria as it currently stands. Therefore, we invite you to submit a revised version of the manuscript that addresses the points raised during the review process.

We look forward to receiving your revised manuscript.

Kind regards,

Jitendra Kumar Singh, PhD

Academic Editor

Journal Requirements:

a. State what role the funders took in the study. If the funders had no role in your study, please state: “The funders had no role in study design, data collection and analysis, decision to publish, or preparation of the manuscript.”

b. If any authors received a salary from any of your funders, please state which authors and which funders.

2. Please provide separate figure files in .tif or .eps format.

4. In the online submission form, you indicated that "We used the Demographic and Health Survey data, which is available from DHS website upon request.". All PLOS journals now require all data underlying the findings described in their manuscript to be freely available to other researchers, either 1. In a public repository, 2. Within the manuscript itself, or 3. Uploaded as supplementary information.

Additional Editor Comments (if provided):

This is overall interesting paper that covers an area of great interest. Factors that are related to prevent low birthweight in South Asia. The authors have made a very good effort overall to describe their findings, considering this is a topic highly quantitative. There are however areas they further need to address.

Were data analyzed clustering at country level? This can highly affect results.

Reviewers' comments:

Reviewer's Responses to Questions

**Comments to the Author**

1. Does this manuscript meet PLOS Global Public Health’s publication criteria? Is the manuscript technically sound, and do the data support the conclusions? The manuscript must describe methodologically and ethically rigorous research with conclusions that are appropriately drawn based on the data presented.

Reviewer #1: Yes

Reviewer #2: Yes

2. Has the statistical analysis been performed appropriately and rigorously?

Reviewer #1: Yes

Reviewer #2: Yes

3. Have the authors made all data underlying the findings in their manuscript fully available (please refer to the Data Availability Statement at the start of the manuscript PDF file)?

Reviewer #1: No

Reviewer #2: Yes

4. Is the manuscript presented in an intelligible fashion and written in standard English?

Reviewer #1: Yes

Reviewer #2: Yes

5. Review Comments to the Author

Reviewer #1: The descriptive statistics sub-section could be more informative. Inference could be deduced from differences in the sample characteristics. Comparing country statistics and exploiting the variance could be a meaningful addition. Further, comparing the size of the parameter estimates between countries and providing plausible reasons while the odds in Nepal, for instance, is different from India, and whether quality and quantity could be explanation.

Reviewer #2: This manuscript uses nationally representative household surveys using Demographic and Household data from six South Asian countries to examine the relationship between quantity and/or quality of womens ANC visit and LBW infants. This is an important body of research given the attention on vulnerable newborns and the research offers important insights for policy.

In general, it would be helpful to provide further comparisons of the differences across the South Asian countries and why these differences may exist in terms of prevalence of LBW infants and/or differences in QoC received by mothers. One smaller addition that would be helpful and benefit the readership would benefit this analysis is to provide a breakdown of LBW infants into moderately LBW, very LBW and Extremely LBW infants to see the prevalence, burden and note if ANC is related to any one of these in particular given the larger sample size.

6. PLOS authors have the option to publish the peer review history of their article (what does this mean?). If published, this will include your full peer review and any attached files.

**Do you want your identity to be public for this peer review?** For information about this choice, including consent withdrawal, please see our Privacy Policy.

Reviewer #1: No

Reviewer #2: No

---

## [Editor Report · Decision Letter 1]

8 May 2023

More is not enough: high quantity and high quality antenatal care are both needed to prevent low birthweight in South Asia

PGPH-D-22-01824R1

Dear Scott,

We are pleased to inform you that your manuscript 'More is not enough: high quantity and high quality antenatal care are both needed to prevent low birthweight in South Asia' has been provisionally accepted for publication in PLOS Global Public Health.

Best regards,

Ramachandran Thiruvengadam, M.D.,

Academic Editor